# Clerodane Diterpenoids from an Edible Plant *Justicia insularis*: Discovery, Cytotoxicity, and Apoptosis Induction in Human Ovarian Cancer Cells

**DOI:** 10.3390/molecules26195933

**Published:** 2021-09-30

**Authors:** Idowu E. Fadayomi, Okiemute R. Johnson-Ajinwo, Elisabete Pires, James McCullagh, Tim D.W. Claridge, Nicholas R. Forsyth, Wen-Wu Li

**Affiliations:** 1School of Pharmacy and Bioengineering, Keele University, Stoke-on-Trent ST4 7QB, UK; i.e.fadayomi@keele.ac.uk (I.E.F.); okiemute_2002@yahoo.co.uk (O.R.J.-A.); n.r.forsyth@keele.ac.uk (N.R.F.); 2Department of Chemistry, University of Oxford, Mansfield Road, Oxford OX1 3TA, UK; elisabete.pires@chem.ox.ac.uk (E.P.); james.mccullagh@chem.ox.ac.uk (J.M.); tim.claridge@chem.ox.ac.uk (T.D.W.C.)

**Keywords:** ovarian cancer, *Justicia insularis*, diterpenoids, cytotoxicity, induction of apoptosis, target prediction

## Abstract

Objectives: The toxicity of chemotherapeutic anticancer drugs is a serious issue in clinics. Drug discovery from edible and medicinal plants represents a promising approach towards finding safer anticancer therapeutics. *Justicia insularis* T. Anderson (Acanthaceae) is an edible and medicinal plant in Nigeria. This study aims to discover cytotoxic compounds from this rarely explored *J. insularis* and investigate their underlying mechanism of action. Methods: The cytotoxicity of the plant extract was evaluated in human ovarian cancer cell lines and normal human ovarian surface epithelia (HOE) cells using a sulforhodamine B assay. Bioassay-guided isolation was carried out using column chromatography including HPLC, and the isolated natural products were characterized using GC-MS, LC-HRMS, and 1D/2D NMR techniques. Induction of apoptosis was evaluated using Caspase 3/7, 8, and 9, and Annexin V and PI based flow cytometry assays. SwissADME and SwissTargetPrediction web tools were used to predict the molecular properties and possible protein targets of identified active compounds. Key finding: The two cytotoxic compounds were identified as clerodane diterpenoids: 16(α/β)-hydroxy-cleroda-3,13(14)*Z*-dien-15,16-olide (**1**) and 16-oxo-cleroda-3,13(14)*E*-dien-15-oic acid (**2**) from the Acanthaceous plant for the first time. Compound **1** was a very abundant compound (0.7% per dry weight of plant material) and was shown to be more potent than compound **2** with IC_50_ values in the micromolar range against OVCAR-4 and OVCAR-8 cancer cells. Compounds **1** and **2** were less cytotoxic to HOE cell line. Both compounds induced apoptosis by increasing caspase 3/7 activities in a concentration dependent manner. Compound **1** further increased caspase 8 and 9 activities and apoptosis cell populations. Compounds **1** and **2** are both drug like, and compound **1** may target various proteins including a kinase. Conclusions: Clerodane diterpenoids (**1** and **2**) in *J. insularis* were identified as cytotoxic to ovarian cancer cells via the induction of apoptosis, providing an abundant and valuable source of hit compounds for the treatment of ovarian cancer.

## 1. Introduction

Ovarian cancer is the most severe of the gynaecological malignancy worldwide, associated with the highest level of lethality due to lack of efficient screening method and early symptoms. Each year, ovarian cancer is diagnosed in about quarter of a million women worldwide, and it stands as the eighth commonest and seventh leading cause of cancer mortality among women, with 140,000 estimated casualties on a yearly basis [1]. At present, the available treatments for ovarian cancer include surgery, radiotherapy, chemotherapy, and immunotherapy. The most common chemotherapy drugs used for ovarian cancer treatment are carboplatin and paclitaxel. Others include cisplatin, gemicitabine, etoposide, topotecan, liposomal doxorubicin, and cyclophosphamide [2,3]. Targeted PARP inhibitors, such as Olaparib (Lynparza) and Niraparib (Zejula), and antibody based drugs, such as bevacizumab (Avastin) and rucaparib (Rubraca), are used with or after other chemotherapy for advanced ovarian cancer treatment [3]. These novel therapies have significantly improved the management of ovarian cancer. However, they have shortcomings, including severe side effects, development of resistance, and high cost. Therefore, the search for new and affordable drugs that could also reduce the adverse effects and overcome the resistant nature of cancer is highly essential.

Natural products from plants or their derivatives represent an important source of anticancer drugs, such as vinblastine, vincristine, topotecan and irinotecan, etoposide, and paclitaxel [2,4]. In particular, phytochemicals from edible and medicinal plants (e.g., isothiocyanates from cruciferous vegetables, sulforaphane from broccoli [5], curcumin from tumeric, genistein from soybean, resveratrol from grapes, and apigenin from various vegetables [6], etc.) are promising sources of anticancer compounds effective in the chemoprevention and treatment of cancer with lower costs and higher safety profiles with a plethora of mechanisms of action [7,8]. Plants in Nigeria have not been extensively explored to discover anti-cancer agents. Previously, we isolated and identified a number of promising cytotoxic alkaloids from several Nigerian medicinal plants [9,10,11,12,13,14,15,16].

*Justicia insularis* T. Anderson (Acanthaceae family) is an annual to perennial edible plant with medical use as digestive, weaning agent, laxative [17,18,19], and nutritional value [20] in Nigeria and across Africa. The *Justicia* is the largest genus with around 600 species, few of which have been studied in recent decades, although arylnaphthalide lignans and triterpenoid glycosides are indicated as the major types of chemical constituents [21]. Aqueous extracts of *J. insularis* leaves were shown to produce estradiol in vitro [17], promote ovarian folliculogenesis and fertility in female rats [19], possess anti-oxidant activity [20], and to benefit the treatment of anaemia [22]. However, the cytotoxic activity and chemical constituents of *J. insularis* have not been characterized. Here, we report the extraction, bioassay-guided purification/isolation, structural identification, cytotoxicity, apoptosis induction evaluation and target prediction in human ovarian cancer cells of the cytotoxic compounds from *J. insularis*.

## 2. Materials and Methods

### 2.1. Reagents

All the chemicals used were of analytical grade. *n*-Butanol, dichloromethane (DCM), ethyl acetate (EA), *n*-hexane, methanol (MeOH), and trichloroacetic acid (TCA) were products of Fischer Scientific, Loughborough, UK. Cell culture media, Roswell Park Memorial Institute (RPMI) 1640, 10% fetal bovine serum (FBS), l-glutamine, PENSTREP (50 μg/mL penicillin/streptomycin), and phosphate buffered saline (PBS) were obtained from Lonza (Basel, Switzerland). Trizma base, trypsin-EDTA solution, glacial acetic acid, dimethyl sulfoxide (DMSO), sulforhodamine B (SRB) sodium salt, trypan blue, and carboplatin were purchased from Sigma Aldrich (St. Louis, MO, USA) and Caspase-Glo 3/7, 8, and 9 assay kits from Promega, Southampton, UK.

### 2.2. Plant Samples

*Justicia insularis* T. Anderson (Acanthaceae) leaves were sourced from Isiokolo in Kokori Town; Region/Local government area: Ethiope East; State: Delta, Nigeria (Latitude: 5°37′52″ N; Longitude: 6°02′06″ E), authenticated by Mr. Alfred Ozioko and deposited at the International Centre for Ethnomedicine and Drug Development (specimen voucher number: INTERCEDD/1590). The plants were pulverized after drying under shade for 7–10 days at 25 °C.

### 2.3. Extraction Procedure for Justicia insularis

The pulverized *J. insularis* leaves (1.0 kg) were macerated in 1000 mL of DCM and 1000 mL of MeOH for 72 h. The mixture was filtered to obtain the DCM/MeOH extract. The residue was further macerated with 1000 mL of methanol for 72 h. The solution of the extract was collected by filtration and repeated two more times within a 24 h maceration period. The residue was there after soaked in 1000 mL of deionized water and filtered after 72 h of maceration. This was also repeated two more times within 24 h each to increase the yield. The DCM/MeOH extract was combined with methanol extract to yield the organic extract, which was dried using a rotary evaporator at <40 °C. The little remaining solvent was further removed using a desiccator. The aqueous extract was frozen at −80 °C for 24 h before being lyophilized to dryness. 

### 2.4. Solvent Partition of Plant Extracts

The organic extract of *J. insularis* (20 g) was further partitioned with three solvents (*n*-hexane, ethyl acetate, and *n*-butanol) as done previously [9,15]. 

### 2.5. Bioassay-Guided Purification of Bioactive Fraction of Justicia insularis

The column was firstly prepared by suspending 50–80 g of silica gel in hexane. The suspended silica gel was poured into the column and allowed to settle with little solvent above the gel. The bioactive hexane or ethyl acetate fraction *J. insularis* was dissolved in hexane and gently transferred to the surface of the gel in the column using Pasteur pipette. The fractions were eluted with 200 mL of *n*-hexane/ethyl acetate in the following ratios (4:1, 3:1, 2:1, 1:1, 1:2, 1:3, and 1:4) consecutively based on the optimum thin layer chromatography profile obtained from a mobile phase of hexane and ethyl acetate combination. The column was finally washed using 100% methanol to obtain the more polar fractions. Ten sub-fractions were obtained using rotary evaporator and desiccator and their ovarian cancer cell growth inhibitory activities were evaluated using a cell growth assay on the OVCAR-4 cell line (Section 2.12). Each of the sub-fractions from the ethyl acetate fraction showed significant anti-cancer activity. However, EA4 was the most active sub-fraction of the ethyl acetate fraction while the least activities were observed in EA9 and EA10 which were eluted with ethyl acetate/methanol and methanol respectively. EA4 was further purified using column chromatography to yield sub-fractions of EA4. The growth inhibitory activity of the sub-fractions of EA4 from column chromatography was evaluated. Sub-fractions EA4-4 and EA4-6 were further purified using semi-preparative high performance liquid chromatography.

### 2.6. Isolation of Compound ***1*** and ***2*** Using High Performance Liquid Chromatography (HPLC)

The various sub-fractions were further assayed for anti-cancer activities and the most significant active sub-fractions were purified further using semi-preparative HPLC. Briefly, semi-preparative HPLC was done using Agilent 1220 LC, USA. The mobile phase used two solvent systems. Solvent A consisted of 100% water and solvent B was 100% methanol. The mobile phase calibration rose from 50% by 50% (A:B) over a period of 25 min to 100% B and kept at 100% for 10 min at a flow rate of 4mL/min at 215 nm on semi-preparative HPLC column (Phenomenex, Cambridgeshire, UK; 5µm particle size: 9.4 × 250 mm). A major fraction and minor fraction eluted at retention times of 22 and 24 min were collected and dried using a rotary evaporator to yield compound **1** (80 mg, 97% purity) and compound **2** (2 mg, 85% purity), respectively. 

### 2.7. Quantification of Compound ***1*** in the Extracts and Plant Materials

The purity of both compounds and the composition of compound **1** in the total plant organic extracts were determined by using analytical HPLC. Stock solutions of the highly pure compound **1** with a series of concentrations (0.125, 0.25, 0.5, 1.0 and 1.5 mg/mL), and organic extracts solutions at 1.0 mg/mL were prepared. Hence, 10 µL of each stock solution and samples were injected into the analytical HPLC system in duplicates. The mobile phase rose from 20% B (A + B) to 100% B over a period of 25 min and kept at 100% for 6 min at 215 nm on an analytical HPLC column (Phenomenex, UK; 5 µm particle size, 4.6 × 250 mm) at a flow rate of 1 mL/min. For the quantification of compound **1** in organic extracts, a linear calibration curve was made by plotting the area under the peak against the different contractions. The percentage of compound **1** in the organic extracts was calculated based on the calibration curve.

### 2.8. Gas Chromatography Mass Spectrometry Analysis

Briefly, 1.0–2.0 mg of the bioactive fractions and isolated compounds of *J. insularis* were dissolved in 200 μL of ethyl acetate and sonicated at <40 °C for 5 min. 1–2 μL of the solution was subsequently injected into gas chromatography mass spectrometry (GC-MS) system consisting of an Agilent 7890 coupled to Agilent MS model 5975C MSD (Agilent Technologies, Cold Spring, NY, USA). The gas chromatography started at 60 °C for 2 min and increased to 300 °C at the rate of 10 °C/min, which was held at 300 °C for 4 min at a constant helium pressure (10 psi). The mass spectra data were acquired in the scan mode in *m/z* range 40–1000.

### 2.9. Liquid Chromatography Mass Spectrometry (LC-MS) Analysis

The purified compounds were analysed by LC-MS to determine their high resolution molecular mass using a Ultimate U3000 ultra-performance liquid chromatography system with a HESI II electrospray ion source on a Q-Exactive Orbitrap mass spectrometer system (Thermo Scientific, Waltham, MA, USA) as described [12].

### 2.10. NMR Spectroscopy

1D and 2D NMR spectra of compounds **1** and **2** were obtained with a Bruker AVII500 NMR spectrometer (Billerica, MA, USA). 1D NMR spectra of the hexane fraction of *J. insularis* were obtained with a Bruker Ascend 400 NMR spectrometer. ACD/Labs 10 Freeware (Advanced Chemistry Development Inc., Toronto, ON, Canada) or Bruker TopSpin 4.1.3 software was used to analyse the NMR Spectra.

### 2.11. Cell Culture

The human ovarian cancer cell lines and normal human ovarian surface epithelial (HOE) cells were used in this study. Ovarian cancer cell lines (OVCAR-4 and OVCAR-8) were products of American tissue culture collection (ATCC). The HOE cells were purchased from Applied Biological Materials (ABM) Inc. (Vancouver, BC, Canada). The Rosewell Park Memorial Institute (RPMI 1640, Lonza) medium was used in the culturing of OVCAR-4, OVCAR-8, and HOE cells. The medium was supplemented with 2 mM glutamine, 10% foetal bovine serum (FBS), and 50 µg/mL penicillin streptomycin. These cells were incubated in a standard humidified incubator) at 37 °C, 5% carbon dioxide (CO_2_) conditions.

### 2.12. Sulforhodamine B Cell Growth Inhibitory Assay

Sulforhodamine B (SRB) assay was used to determine the inhibition of cell proliferation by the studied compounds and plant extracts [9,10,15]. Plant extracts/fractions (100 mg/mL) and pure compounds (20 mM) were prepared in DMSO. The 0.2% DMSO in growth media was added to the cells as vehicle-treated cells (negative control) while carboplatin was used as positive control. In the SRB assay, OVCAR-4, OVCAR-8, and HOE cells were seeded in 80 µL growth medium per each well in 96 well plates. OVCAR-4 and HOE were seeded at a density of 5000 cells per well while OVCAR-8 was seeded at a density of 2000 cells per well. The seeded plates were incubated for 24 h, after which 20 μL of plant extracts and natural compounds 1000 µg/mL and 200 µM (and their serial two-fold dilutions), respectively, were added at the indicated concentrations. The cell cultures were incubated at 37 °C under 5% CO_2_ for 72 h in a humidified atmosphere. 

After 72 h, the medium was decanted, and the cells fixed with 0.1 mL of 10% TCA on ice for 30 min before drying. The cells were stained with 0.4% SRB, washed with 1% acetic acid three times and dried. Then, 0.1 mL of Tris-base (10 mM) was added to the plates to solubilise the protein-bound SRB dye. The absorbance at 570 nm was measured using a spectroscopic plate reader (Multi-mode microplate reader BioTEK Synergy 2, Winooski, VT, USA). The data were analysed by non-linear regression to fit a 4-parameter sigmoidal dose–response curve to determine IC_50_ values using GraphPad PRISM 6.0 software, Inc. (San Diego, CA, USA).

### 2.13. Apoptosis Detection Using Caspase-Glo 3/7, 8 and 9 Activity Assay

Caspase 3/7 activities was measured using assay kits caspase-Glo 3/7 (Promega Corp., Madison, WI, USA) on a 96-well microplate. Briefly, the cells (OVCAR-4 and OVCAR-8) were seeded in 96 well plates at a cell density of 5000 cells/well in 80 μL growth media and exposed to 10, 20 and/or 30 μM of the natural compounds after 24 h incubation. After 48 h exposure to compound treatments, 25 μL of Caspase 3/7 Glo-reagent was added, and the cells were incubated in the dark at room temperature for 30 min on a gentle rocker. The luminescence was measured at 570 nm by a BioTEK Synergy microplate reader (USA). A similar procedure was followed for caspase 8 and caspase 9 activity.

### 2.14. Evaluation of Early and Late Apoptosis Using Flow Cytometry

The in vitro method of fluorescence-activated cell sorting (FACS) by Annexin V and propidium iodide (PI) staining was used to detect the change of cell population as reported [10,23]. OVCAR-8 cells were seeded in 12 well plates at a density of 2 × 10^5^ cells per well in 1 mL of growth media and incubated for 24 h before treatment with the tested compound and positive control. After treatment for 48 h, media were decanted into 15 mL tubes, and cell pellets were collected into the same tubes by trypsinisation, the cells were centrifuged at 150× *g* for 3 min and re-suspended into 1 mL growth media, which was transferred into sterile 2 mL Eppendorf tube, centrifuged at 300× *g* for 5 min at 4 °C. The media were aspirated, and the pellet was washed in cold PBS. The cells were centrifuged under the same conditions and the supernatant aspirated. Cells were washed in 500 µL annexin-V binding buffer and centrifuged at 300× *g* for 10 min, annexin-V binding buffer was aspirated, and the cell pellet was treated with 10 µL of annexin V-FITC in 100 µL of binding buffer. The cells were thoroughly mixed and incubated in the dark at room temperature for 15 min. After incubation, cells were washed in binding buffer and centrifuged at 300× *g* for 10 min. The buffer was aspirated, and cells were suspended in 500 µL binding buffer and subsequently 5 µL of PI was added for flow cytometry analysis.

### 2.15. Bioinformatic Analysis

The molecular properties of compounds **1** and **2** were determined or predicted through the SwissADME website tool [24,25], and their molecular targets predicted through SwissTargetPrediction (Swiss Institute of Bioinformatics, University of Lausanne, Lausanne, Switzerland) web tool [26,27].

### 2.16. Statistical Analysis

The IC_50_s were obtained from at least three repeated experiments. The mean IC_50_ was calculated, and the standard error of mean (SEM) was determined. Furthermore, a one way analysis of variance (ANOVA) and student *t* test were used to test if the difference in the mean of control and treatments and mean of treatment at different concentration were significant. A post hoc Dunnett test was used to determine which of the treatments was significant to the control while a Tukey test was used to determine which of the concentrations of a particular treatment were significant using GraphPad prism 6.

## 3. Results

### 3.1. Bioassay-Guided Isolation of Diterpenoids from J. insularis

The bioassay-guided fractionation and isolation of two diterpenoids from extracts of *J. insularis* are illustrated (Appendix A). Both organic (21.0 g, 2.1% yield) and aqueous extracts of *J. insularis* were obtained. The organic extracts were partitioned with solvents and the weights and yields of the solid fraction recovered are 4.6 g (25%), 6.0 g (33.3%), 2.5 g (13.9%), and 3.0 g (16.7%) for *n*-hexane, ethyl acetate (EA), *n*-butanol, and aqueous fractions respectively. 

The organic extract showed stronger cytotoxicity against OVCAR-4 and OVCAR-8 cell lines (IC_50_ < 30 μg/mL) than the aqueous extract *J. insularis* using SRB cell growth assay. Furthermore, that n-hexane and ethyl acetate fractions derived from the organic extract are the most active fractions with IC_50_ less than 20 μg/mL (Appendix A, Appendix A). Analytical HPLC analysis of both n-hexane and ethyl acetate fractions showed similar patterns of compounds. The more abundant ethyl acetate fraction of *J. insularis* was focused on and subjected to silica gel column chromatography. Ten sub-fractions were obtained, and their in vitro ovarian cancer growth inhibitory activities were evaluated (Appendix A). The most active sub-fraction (EA4) was further purified using reversed-phase HPLC to yield compounds **1** and **2** with high purity (Appendix A).

### 3.2. Chemical Identification of the Isolated Bioactive Compounds of J. insularis

The molecular formula of **1** was determined as C_20_H_30_O_3_ based on the observed molar mass of compound **1** (found, 318.2123 Da) by LC-HR-MS (Appendix A). Compound **1** (Figure 1) was identified as 1:1 mixture of 16-hydroxy epimers (α and β) of 16-hydroxy-cleroda-3,13(14)*Z*-dien-15,16-olide based on GC-MS (Appendix A), ^1^H, ^13^C-NMR and HSQC analysis (Appendix A, Appendix A) and a comparison with literature data [28,29,30]. ^13^C-NMR spectrum of the hexane fraction (Appendix A) showed the presence of epimers of compound **1** before silica gel chromatography and HPLC purification which may cause isomerization, so the epimers of compound **1** are the natural products in *J. inuslaris*. Compound **1** showed a single peak in analytical HPLC chromatogram indicating a purity of 97% (Appendix A). Surprisingly, three major peaks appeared on the GC-MS chromatogram (Appendix A), the mass spectrum of a peak at retention time 18.45 min is consistent with the mass data of compound **1** [24]. The other observed compounds, 1a and 1b (Appendix A), were determined to be the thermal degradation products of compound **1** due to the presence of a γ–hydroxy unsaturated 5-membered lactone moiety (Figure 1) under high temperature conditions (60 °C–300 °C) for 30 min of GC-MS. To support this, the exposure of compound **1** at 200 °C for 0.5 h followed by HPLC analysis indicated the formation of new products (likely including **1a** and **1b**), whose structures remain to be determined. Furthermore, the percentage of compound **1** in organic extracts was determined to be 34%. Thus, the composition of compound **1** in the plant material is 0.7% (dry weight), a very abundant secondary metabolite in the leave of *J. insularis*.

Compound **2** (Figure 1) has the same molecular formula C_20_H_30_O_3_ as compound **1** by LC-HRMS data (Appendix A), which was further identified as 16-oxo-cleroda-3,13(14)*E*-dien-15-oic acid based on ^1^H-NMR, ^13^C-NMR and HSQC analysis (Appendix A) and a comparison with literature data (Appendix A) [28,29].

### 3.3. In Vitro Cytotoxicity of Compounds ***1*** and ***2***

Compound **1** (IC_50_ = 4–6 μM) shows greater potency than compound **2** (IC_50_ = 12–17 μM) and a positive control carboplatin (IC_50_ = 8–18 μM) (Table 1) against OVCAR-4 and OVCAR-8 cells (Figure 2). Both compounds **1** and **2** demonstrate less cytotoxic activity against HOE cells (Table 1).

### 3.4. Apoptosis Study 

To investigate whether the significant decrease in cell viability by compound **1** and **2** was due to apoptosis, the level of caspase 3/7, or caspase 8 and 9 activation by these compounds were measured. Compound **1** and **2** significantly increased caspase 3/7 activities in OVCAR-4 and OVCAR-8 cells when compared with control using one-way analysis of variance (ANOVA) (Figure 3A,B). Compound **1** further increased caspase 8 and 9 activities in OVCAR-8 (Figure 3C,D). These results suggest that the significant decrease in cell viability induced by these compounds was likely due to apoptosis.

Furthermore, annexin-V and PI assay results also show concentration-dependent and significant increase of percentage of early and late apoptosis induced by compound **1** (at 5, 10, and 20 µM) after 48 h treatment (Figure 4), similar to the positive control carboplatin same concentrations. 

### 3.5. Bioinformatic Analysis

Compounds **1** and **2** have the same molecular formula, with **1** having a more rigid structure than **2**. Both compounds are druglike, obeying the Lipinski rule (Appendix A). The possible targets of compound **1** are predicted to be mainly kinase (e.g., Ribosomal protein S6 kinase alpha 5), primary active transporters (e.g., potassium-transporting ATPase alpha chain 2), and nuclear receptors (e.g., glucocorticoid receptor) (Appendix A, Appendix A), while the possible targets of compound **2** are predicted to be nuclear receptors (e.g., estrogen receptor beta), oxidoreductases (e.g., steroid 5-alphareductase 2), and phosphatases (e.g., protein-tyrosine phosphatase 1B) (Appendix A, Appendix A), although with low probability (<25%). 

## 4. Discussion

The ethnopharmacological use and the anti-cancer activity of *Justicia* species [31,32,33] inspired us to investigate the cytotoxic activities of *J. insularis* from Nigeria against ovarian cancer cells. In this study, two clerodane diterpenoid compounds (**1** and **2**) were revealed to be the cytotoxic compounds in *J. insularis* for the first time. The very abundant compound **1** in the plant extracts showed higher potency with IC_50_ values < 6 µM against the two ovarian cancer cell lines studied and greater SI compared to those of a standard chemotherapeutic drug carboplatin for ovarian cancer treatment. This makes compound **1** an interesting hit compound. The cytotoxic activity of compound **1** is likely associated with the more rigid α,β-unsaturated γ-lactone moiety in the clerodane diterpenoid, whereas compound **2** with an open form and more rotational bonds (Appendix A) demonstrates less cytotoxicity [34]. Compound **1** was found to be thermally unstable at a high temperature, and further isolation, identification, and testing the cytotoxicity of those degradation compounds (Appendix A) would be interesting. Compound **1** was found to be a mixture of epimers of 16α (S) and 16β (R) forms (1:1) based on the analysis of NMR spectra of the isolated compound **1** and the hexane fraction before purification (Appendix A). The single epimer 16α (S) form of compound **1** was previously isolated from *P. longifolia* [28], *P. barnesii* [35], and *P. simiarum* [36] (the Annonaceae family). A mixture of 16S and 16R epimers (1:1) was also found in *P. longifolia* [24] and chemically synthesized [30]. Compound **2** was also present in *P. longifolia* [28,29]. Previously, phytochemical studies of *Justicia* species disclosed the presence of cytotoxic lignans, such as justicidin A [37,38] and 6′-hydroxy justicidin A [32] from *J. procumbens* and triacontanoic ester of 5-hydroxy-justisolin from *J. simplex* [31]. Chemical analysis of leave extract of *J. insularis* indicated the presence of abundant iron [20] and hemoglobin, which might explain the observed benefit for anaemia [22]. Our study indicates the clerodane diterpenoids, such as compounds **1** (abundant) and **2**, are present in the family of Acanthaceae, providing a new source of these interesting compounds as hit compounds for anticancer drug development.

Cancer cells are generally known for their characteristic features of escaping programmed cell death (apoptosis), which is a mechanism that maintains the cell population and defense against damaged cells [39]. To further investigate the route of anticancer activity for compound **1** and **2**, their roles in the induction of apoptosis were evaluated. The caspase 3/7 activity of compounds **1** and **2** was evaluated, and the results showed that the cell death induced by compounds **1** and **2** was via the activation of caspase 3/7 which are apoptosis executioners. Furthermore, compound **1** activated both caspase 8 and 9, which indicates the involvement of both extrinsic and intrinsic pathway [39]. Compound **2** was not further investigated due to the lesser activity and limited quantity isolated. The pro-apoptotic activity of compound **1** was further verified using annexin V-FITC and PI staining, which analysed the apoptotic markers, i.e., phosphatidylserine residues, on the cell surfaces and DNA fragments in the nucleus, respectively [23]. The percentages of both early and late apoptotic cells caused by compound **1** are similar to those of carboplatin. Further bioinformatic analysis of the potential protein targets of compounds **1** and **2** supports the observed greater cytotoxicity and apoptotic activity of compound **1** than **2**, because compound **1** may more likely target kinases which are essential for cancer cell initiation and proliferation [40]. The cytotoxicity and induction of apoptosis of compound **1** were previously observed in other cancers, such as leukaemia HL-60 [41], CML K562 [42,43], oral squamous cell carcinoma cancer [44], human renal carcinoma [45], renal cell carcinoma [46], T24 bladder cancer [47], and breast cancer [48] cells. Specifically, compound **1** induces the expression of PRC2 enzyme complex in CML K562 cells [42] and deregulates phosphoinositide-3 kinase (PI3K) and Aurora kinase B activities [43]. Compound **1** was also found to be involved with Akt, mTOR, and MEK-ERK pathways in renal carcinoma cells [45] or the inactivation of EGFR-related pathways in bladder cancer cells [47]. Our study is the first report showing their cytotoxicity in ovarian cancer cells via the induction of apoptosis via both intrinsic and extrinsic pathways, which is consistent with its cytotoxicity found in other cancers.

Besides the cytotoxic activity of these diterpenoids, 16α-hydroxy-cleroda-3,13(14)*Z*-dien-15,16-olide (**1**) was also demonstrated to be an orally active anti-leishmanial and non-cytotoxic agent [49], a HMG-CoA reductase inhibitor [50], and a dual inhibitor of COX/5-LOX with potential in the treatment of inflammatory conditions [51,52]. Compound **2** possesses anti-biofilm activity against methicillin resistant *Staphylococcus aureus* and *Streptococcus mutans* [53]. 

Diterpenoids are a large group of natural products with diverse structures and biological activities including anticancer activity [34,54]. One of the well-known and approved anticancer diterpenoid drug from plants is paclitaxel, which showed superior potency against ovarian cancer cells with IC_50_s at nanomolar range through stabilizing the tubulin structure of cancer cells [55]. However, it also caused severe side effects among patients because of its poor selectivity to cancer cells. So far, more than hundreds of important diterpenoids, including triptolide, oridonin, and andrographolide, have been discovered and shown in vitro and/or in vivo cytotoxicity with moderate and strong potency [54]. Specifically, clerodane diterpenoids were isolated and showed cytotoxicity to various cancer cell lines [34]. For example, caseamembrins A–F from *Casearia membranacea* showed cytotoxicity to human prostate cancer PC-3 cells with IC_50_ at the µM range with either intrinsic or extrinsic apoptotic pathways [56]. *Ent*-clerodane diterpenoids from *Scutellaria barbata* D. Don. (Labiatae) showed cytotoxicity against KB oral epidermoid carcinoma, HONE-1 nasopharyngeal, and HT29 colorectal carcinoma cells with IC_50_ values in the range of 3.1–7.2 µM [57]. Kurzipene B from the leaves of *Casearia kurzii* showed an IC_50_ value of 5.3 µM against Hela cells, induced apoptosis, and arrested the cell cycle at the G_0_/G_1_ stage [58]. The activity of the two compounds found in *J. insularis* is comparable to clerodane diterpenoids from other plants.

Further work on the experimental identification of molecular targets, structure–activity relationships, efficacy, and safety of diterpenoids in *J. insularis* remain under investigation in in vitro and animal xenograft models, which may provide a safer, more sustainable, and affordable anticancer drug. 

## 5. Conclusions

This study shows that the extract of the edible *J. insularis* leaves demonstrate significant cytotoxic activities against ovarian cancer cell lines, providing a new and abundant potential source of anticancer agents. Two diterpenoids, 16α/β-hydroxy-cleroda-3,13(14)*Z*-dien-15,16-olide (**1**) and 16-oxo-cleroda-3,13(14)*E*-dien-15-oic acid (**2**), were isolated and identified as the cytotoxic compounds through bioassay-guided fractionation from the Acanthaceae family. Compound **1** showed greater selectivity towards in vitro cancer cells over normal cells. Compound **1** was established to induce apoptosis through both intrinsic and extrinsic pathways, which warrants the further investigation of compound **1** as an anticancer agent from an edible plant as with the potential of lower toxicity. 

## Figures and Tables

**Figure 1 molecules-26-05933-f001:**
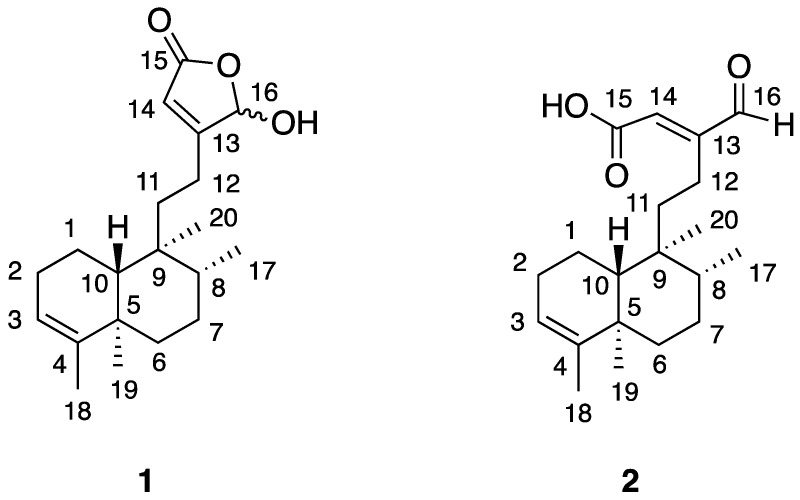
Chemical structure of compound **1**, 16(α/β)-hydroxy-cleroda-3,13 (14)*Z*-dien-15,16-olide, and compound **2**, 16-oxo-cleroda-3,13(14)*E*-dien-15-oic acid.

**Figure 2 molecules-26-05933-f002:**
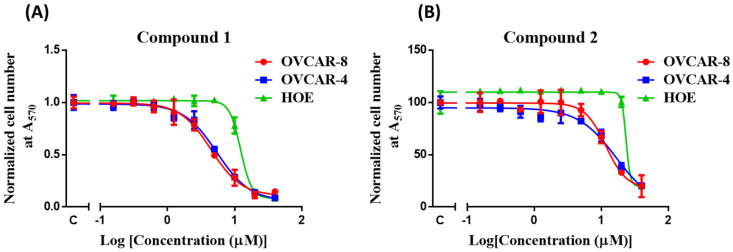
Mean concentration-response curve of compound **1** (**A**) and **2** (**B**) in OVCAR-4 and OVCAR-8 ovarian cancer cells and HOE. IC_50_ values determined are listed in Table 1.

**Figure 3 molecules-26-05933-f003:**
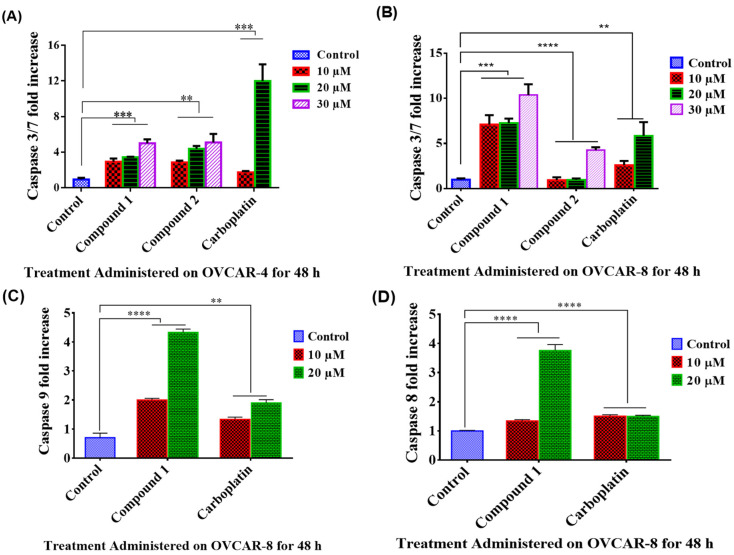
Caspase activities of compound **1** and **2** in ovarian cancer cells. Caspase 3/7 activities of isolated compound **1** and **2** in OVCAR-4 (**A**) and OVCAR-8 (**B**) cells; and caspase 8 activity (**C**) and caspase 9 activity (**D**) of compound **1** in OVCAR-8 cells. Carboplatin was used as positive control. The fold increase in caspase activities induced by compound **1** and positive control were compared with the negative control using one-way ANOVA with Dunnett’s multiple comparison test. Significant difference between treatment and control is denoted with asterisk (*) and student *t* test was used to test for concentration dependent activity.

**Figure 4 molecules-26-05933-f004:**
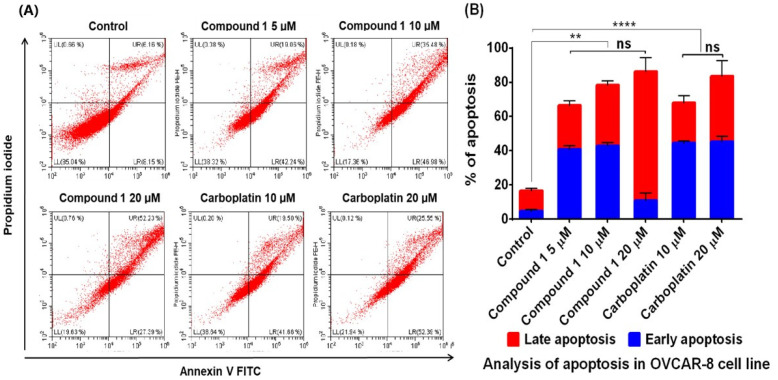
Evaluation of apoptotic activities of compound **1** (5, 10 and 20 µM) and carboplatin (10 and 20 µM) on OVCAR-8 cells using annexin V-FITC and propidium iodide (PI) staining analysed with flow cytometry. (**A**) Representative flow cytometry graphs of OVCAR-8 cell line. Lower left (LL), upper left (UL), lower right (LR) and upper right (UR) represent live cells, necrotic cells, cells in early apoptosis and cells in late apoptosis respectively. (**B**) Mean percentage of apoptotic cell populations. The data represent the mean ± SD of three repeats. The significant different between control and treatment is denoted with asterisk (*), while no significant different is denoted with (ns).

**Table 1 molecules-26-05933-t001:** The growth inhibitory activities of isolated compounds (**1** and **2**) from *J. insularis* in OVCAR-4, OVCAR-8 cancer cell lines and HOE cells. The selectivity index (SI) (the ratio of IC_50_ against HOE cells to IC_50_ against OVCAR-8) are indicated.

Compounds	OVCAR-4 (μM)	OVCAR-8 (μM)	HOE (μM)	SI against OVCAR-8
**1**	5.7 ± 0.3(1.8 μg/mL)	4.4 ± 0.2 (1.4 μg/mL)	12.1 ± 0.1 (3.9 μg/mL)	3
**2**	16.6 ± 2.8 (5.3 μg/mL)	11.8 ± 0.5 (3.8 μg/mL)	22.8 ± 0.7 (7.3 μg/mL)	2
Carboplatin	17.6 ± 4.6	8.2 ± 2.2	13.0 ± 3.7	1.6

## Data Availability

Data available within the article or Appendix A.

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
