# Peer review of "Clerodane Diterpenoids from an Edible Plant Justicia insularis: Discovery, Cytotoxicity, and Apoptosis Induction in Human Ovarian Cancer Cells"

_molecules, 2021, doi:10.3390/molecules26195933_

Round 1

Reviewer 1 Report

The authors have found that an edible and medicinal plant Justicia insularis growing in Nigeria is an abundant source of two clerodane diterpenoids (γ-lactone and its open form isomer), which biological activities, e.g. anti-leishmanial, anti-inflammatory, HMG-CoA reductase inhibitory, have been previously reported. The clerodane diterpenoidal lactone have also shown cytotoxicity and induction of apoptosis in cancer cell lines such as leukaemia HL-60, CML K56, oral squamous cell carcinoma, human renal carcinoma, bladder cancer, and breast cancer. Now this compound has been identified as cytotoxic to ovarian cancer cells via induction of apoptosis. The paper is sound and I recommend it for publication in Molecules essentially as is.

The authors may consider to explain either in the text or in SI, how did they establish the 1 : 1 ratio of epimers at C16 in compound 1. Is the natural product an epimeric mixture or the epimerization occurs during its isolation?

Author Response

Thanks for the good point.

The ratio of the 16a and 16b form of compound 1 is determined to be 1:1 by calculating the ratio of the integration value of the carbon peak at 116.94 ppm (C-14, 16alpha form) to that of the peak at 116.97 ppm (C-14, 16beta form).

Additional NMR of the hexane fraction (Figure S6, C)  has been done and indicated that the epimers of compound 1 were present in the hexane fraction before purification process which may cause isomerization, so epimers of compound 1 are natural products in J. inuslaris.  

These have been added to the Manuscript and Supplementary Material, Figure S6 legend.

Reviewer 2 Report

After reading the manuscript entitled: Clerodane diterpenoids from an edible plant Justicia insularis: discovery, cytotoxicity, and apoptosis induction in human ovarian cancer cells, follow my considerations.

Page 3, line 1: On sub-item: 2.3. Extraction procedure for Justicia insularis: “...to yield the organic extract (21.0 g, 2.1% yield)...”.

Page 3, lines 8 and 9: On sub-item: 2.4. Solvent partition of plant extracts “... and yield of the solid fraction recovered are 4.6g (25%), 6.0g (33.3%), 2.5g (13.9%) and 3.0g (16.7%) for n-hexane,...”

Comments: I kindly ask the authors to include the yield data in the Results section

Page 3, lines 23, 24, 25 and 26: On sub-item: 2.5. Bioassay-guided purification of bioactive fraction of Justicia insularis  “... However, EA4 was the most active sub-fraction of the ethyl acetate fraction while the least activities were observed in EA9 and EA10 which...”....

“...EA4 which was the most active fraction was further...”

Comments: I kindly ask the authors to include the response of anti-cancer activity in the Results section.

Page 3, lines 39 and 40: On sub-item: 2.6. Isolation of compound 1 and 2 using high performance liquid chromatography (HPLC)  “....to yield compound 1 (80 mg, 97 % purity) and compound 2 (2 mg, 85 % purity)...”

Comments: I kindly ask the authors to include the yield data in the Results section

In the “Supplementary Materials”: In Table S3:

Comments: I kindly ask the authors to include the H-6 hydrogen spectral data for both compounds (compound 1 and 2). For hydrogens 11 (compound 1) and 12 (compound 2), the values of geminal coupling constants need to be equal. For hydrogens 12 in compound 1, what does Hα , 1Hβ , 1Hα  1Hβ mean? The values of geminal coupling constants need to be equal too. For hydrogens 17 in compound 2, what is the value of the coupling constant for the doublet?

Finally, this manuscript is interesting and I suggest to accept it considering some minor corrections suggested above.

Author Response

Thanks for positive comments.

The suggested results have been replaced in the Results section.

The requested NMR data and explanation of alpha and beta form in the Table S3 of the Supplementary Material have been added and clarified.

Please see the revised files highlighted in yellow colour.

Reviewer 3 Report

Authors of “Clerodane diterpenoids from an edible plant Justicia insularis: discovery, cytotoxicity, and apoptosis induction in human ovarian cancer cells” made an interesting work which can be published in Molecules after doing some corrections.

They obtained two compounds with antiproliferative activity. In fact, they are structurally related. Nevertheless, they did not discuss the difference in biological activity by means of molecular descriptors. These molecular descriptors can be calculated by many web servers like molinspiration or swissADME.

In addition, by using these servers they may enhance the discussion of the druggability of their compounds. Even they can search for possible biological targets by using some servers, like PASS online, binding database, etc.

Author Response

Thank you for excellent suggestions. 

We have used SwissADME and SwissTargetPrediction web tools to provide the molecular properties and possible targets of compounds 1 and 2.

These results have been described in the manuscript and supplementary materials.

This manuscript is a resubmission of an earlier submission. The following is a list of the peer review reports and author responses from that submission.